# Exosomal microRNA Panels for Detecting Early-Stage Non-Small Cell Lung Cancer

**DOI:** 10.3390/diagnostics15212735

**Published:** 2025-10-28

**Authors:** Young Jun Kim, Da Hyun Kang, Hyunmin Cho, Chaeuk Chung, Jeong Eun Lee, Yong-Beom Shin

**Affiliations:** 1BioNano Health Guard Research Center (H-GUARD), Daejeon 34109, Republic of Korea; btsmzt@naver.com (Y.J.K.); hyunmin@h-guard.re.kr (H.C.); 2Department of Internal Medicine, College of Medicine, Chungnam National University, Daejeon 35015, Republic of Korea; ibelieveu113@cnuh.co.kr (D.H.K.); universe7903@gmail.com (C.C.); 3Bionanotechnology Research Center, Korea Research Institute of Bioscience and Biotechnology (KRIBB), Daejeon 34141, Republic of Korea

**Keywords:** microRNA, non-small cell lung cancer, up-down ratio, early-diagnosis

## Abstract

**Background**: Early diagnosis of lung cancer requires lung nodule biopsies, which can lead to severe complications. This study aimed to identify optimized panels of exosomal microRNAs (miRNAs) for non-invasive diagnosis of early-stage non-small cell lung cancer (NSCLC). **Materials and Methods**: This study comprised four phases: discovery, validation, optimization, and confirmation. In the discovery phase, next-generation sequencing profiled 2656 exosomal miRNAs in serum samples (*n* = 76) from patients with benign lung nodules and stage-specific NSCLC. The validation phase used qPCR to analyze selected miRNAs in serum samples (*n* = 75). The optimization phase employed a self-devised diagnostic platform, the “up-down ratio (UDR),” to identify miRNA panels. The confirmation phase involved miRNA–target gene interaction and enrichment analyses. **Results**: The discovery phase identified 15 candidate miRNAs, of which six were validated by qPCR: miR-1976, miR-150-5p, miR-301b-3p, miR-369-3p, miR-497-5p, and miR-610. UDR platform identified a panel of four miRNAs optimized for early detection of NSCLC with ROC over 0.93. Bioinformatics analysis revealed 20 target genes, with *VEGFA*, *BCL2*, and *PTEN* showing strong interactions with the miRNAs, particularly with miR-150-5p, miR-205-5p, miR-1976, miR-301b-3p, and miR-497-5p. **Conclusions**: This four-phase study suggests that exosomal miRNA panels have potential diagnostic value for early-stage lung cancer. The UDR platform enabled the selection of a four-miRNA panel (miR-150-5p, miR-301b-3p, miR-369-3p, and miR-497-5p), with bioinformatics analyses providing supportive evidence.

## 1. Introduction

Lung cancer (LC) is a leading cause of cancer-related deaths, accounting for 1.79 million deaths worldwide in 2020. Among different types of LCs, non-small cell LC (NSCLC) accounts for approximately 85% of all LC incidents [1]. In general, the 5-year survival rate of patients with LC is 15% [2], whereas that of the patients with stage I NSCLC can reach up to 80% [3]. As almost all patients with stage I LC are asymptomatic, detecting lung cancer at an early stage is difficult. 75% of patients with LC are diagnosed at stage III or IV, in which case the overall survival rate decreases to 37% and 6%, respectively [3].

Clinically, detection of LC is processed through imaging techniques, including chest X-ray and low-dose computed tomography (LDCT). LDCT is currently recommended for individuals at high risk [4]. However, these imaging techniques present limitations in performance associated with low specificity and accessibility to facilities [5]. In addition, although a biopsy of lung nodules is essential for LC diagnosis, lung biopsy is relatively invasive compared with other tissues, such as the breast and thyroid, and sometimes causes serious complications, including massive hemoptysis or fatal pneumothorax [6,7,8]. Therefore, when a small nodule revealing indeterminate features is observed on computed tomography (CT), clinicians experience difficulty deciding whether to perform a biopsy or observe a short-term follow-up. Non-invasive biomarkers, whether alone or combined with imaging techniques, are expected to improve sensitivity and specificity in diagnosing early-stage LC. Many different protein-based cancer biomarkers have been reported, including alphas-fetoprotein in liver cancer [9], cancer antigen 125 (CA 125) in ovarian cancer [10], carbohydrate antigen 19-9 (CA 19-9) in pancreatic cancer [11], carcinoma embryonic antigen in colorectal cancer [12], and prostate-specific antigen in prostate cancer [13]. A few hundred protein biomarkers have been developed for LC, among which tumor-associated antigen, tumor-associated autoantibodies, and exosomal proteins are the most widely studied. However, all those cancer-targeting protein biomarkers lack the sensitivity and specificity required for diagnosing early-stage LC [14].

MicroRNAs (miRNAs), classified as non-coding small RNA molecules, comprise approximately 22 nucleotides. Over 2500 miRNAs of human origin have been registered in miRBase. Considering their role as modulators of gene expression, dysregulation of miRNAs in human biological fluids is a biomarker for the pathological status of various diseases, including cancers [14,15,16]. From the viewpoint of biomarkers, miRNAs are considered advantageous because they are non-invasively collectible and detectable and can retain stability in a clinically manageable environment owing to extracellular vesicles or their complexation with proteins and Argonaute 2 [14,17,18]. From an analytical perspective, miRNAs can be measured and studied dependably using real-time quantitative polymerase chain reaction (qPCR), microarray, and next-generation sequencing (NGS). In addition, miRNAs with over 2500 species can be used as multiple biomarkers, improving detection performance with increased sensitivity and specificity in comparison to single use of biomarkers, especially for such complex diseases as cancers [19,20,21].

Circulating cell-free miRNAs remain stable despite RNase activity, as described above; however, those not associated with vesicles and exosomes may be less stable [22]. A recent review comparing exosomal and non-exosomal miRNAs recommended exosomal miRNAs for biomarker studies [23]. In this study, we aimed to identify exosomal miRNA panels with high diagnostic potential for early-stage NSCLC using a four-phase approach and a novel optimization platform.

## 2. Materials and Methods

### 2.1. Study Design

We conducted a four-phase study to identify circulating miRNA panels for the detection of early-stage NSCLC: discovery, validation, optimization, and confirmation (Figure 1). In the discovery phase, next-generation sequencing (NGS) was performed on serum samples that passed quality control (QC) testing (*n* = 76, Appendix A). In the validation phase, qPCR analysis was performed on an independent cohort (*n* = 75), resulting in the identification of six individual miRNAs that showed significant differential expression between stage I+II NSCLC and benign cohorts. In the optimization phase, a novel diagnostic platform, the “up-down ratio (UDR),” was introduced, in which the average expression level of the upregulated miRNAs was divided by that of downregulated ones to establish optimal diagnostic panels (Equation (1)). Finally, in the confirmation phase, bioinformatics analysis was conducted to confirm the diagnostic reliability of the selected miRNAs.(1)$$UDR=\{(\sum_{j=1}^{N}UE)/N\}/\{(\sum_{j=1}^{M}DE)/M)\}$$
UDR: up-down ratioUE: expression level of up-regulated miRNAsN: number of up-regulated miRNAsDE: expression level of down-regulated miRNAsM: number of down-regulated miRNAs


### 2.2. Patient Cohorts and Sample Collection

A total of 200 serum samples were prospectively collected at Chungnam National University Hospital in South Korea. Among them, 137 samples were obtained from patients with histologically confirmed NSCLC (38 stage I, 26 stage II, 38 stage III, and 35 stage IV), and 63 samples were obtained from patients with benign pulmonary nodules. In this study, “early-stage NSCLC” was defined as stage I–II disease, consistent with widely accepted classifications. The discovery cohort consisted of 125 samples collected between February and August 2018, and the validation cohort consisted of 75 samples collected between September 2018 and April 2019. Patients were eligible if they had histologically confirmed stage I–IV NSCLC or benign pulmonary disease and provided written informed consent. The benign pulmonary nodules were all pathologically confirmed by biopsy or surgical resection to ensure their non-malignant nature. Patients were excluded if their diagnosis or staging was uncertain, if clinical information was insufficient, or if the collected serum samples were of inadequate quality. All serum samples were collected at diagnosis, prior to any cancer treatment, including chemotherapy, radiotherapy, or surgery. The study protocol was approved by the Institutional Review Board of Chungnam National University Hospital (IRB No. 2018-01-059).

### 2.3. Sample Preparation

Blood samples (5 mL) were collected via venous puncture into serum separator tubes and centrifuged at 3000 rpm for 10 min at 4 °C. Serum was separated within 2 h of collection, aliquoted, and immediately stored at −80 °C until use.

### 2.4. Exosome Isolation

Following manufacturer’s instructions, exosomes were isolated from serum samples using miRCURY^®^ Exosome Serum/Plasma Kit (cat. 76603, Qiagen, Hilden, Germany). Briefly, 1 mL of serum sample was centrifuged at 3000× *g* for 10 min to obtain cell-free serum. Precipitation Buffer A (200 μL) was gently mixed with the cell-free serum and incubated for 1 h at 4 °C. Exosomes were pelleted by centrifugation at 1500× *g* for 30 min at 20 °C, and the resulting pellet was resuspended in 270 μL of Resuspension Buffer by vortexing.

### 2.5. RNA Extraction

RNA was extracted from the exosome suspension using the miRNeasy Serum/Plasma Kit (Qiagen), following the manufacturer’s protocol. Briefly, 1 mL of QIAzol Lysis Reagent was mixed with 200 μL of exosome suspension, and the mixture was incubated at atmospheric temperature for 5 min. An exogenous spike-in control (3.5 μL of cel-miR-2-3p, 1.6 × 10^8^ copies/μL) was added to the lysate. In two separate steps, RNA precipitation was carried out with 200 μL of chloroform and 900 μL of 100% ethanol. Then, 700 μL of the sample was added to the RNeasy MinElute spin column (Qiagen, Hilden, Germany) and centrifuged at 11,000× *g* at atmospheric temperature for 15 s. The column was washed with 700 μL of RWT and 500 μL of RPE buffer. The solution was centrifuged at 11,000× *g* at room temperature for 15 s, followed by RNA precipitation with 500 μL of 80% ethanol. RNA was eluted from the column with 14 μL RNase-free water.

### 2.6. RNA Quality Check

RNA quantity and integrity were assessed using the Quant-IT miRNA assay kit (Thermo Fisher Scientific, Waltham, MA, USA), Quantus TM Fluorometer (Promega. Madison, WI, USA) and Agilent 2100 Bioanalyzer (Agilent Technologies, Santa Clara, CA, USA). Samples were preliminarily categorized into four quality levels (bad, maybe OK, good, and fairly good) based on total RNA amount (10 ng) and 10 fluorescent units (Appendix A). Only samples that successfully passed library construction were applied for NGS analysis.

### 2.7. NGS Analysis of miRNAs

SMARTer smRNA-Seq Kit (Illumina; Takara Bio, Shiga, Japan) was used for library preparation. Library sequencing was performed using polyadenylation, complementary DNA (cDNA) synthesis, and PCR. Library quality was analyzed with Agilent 2100 Bioanalyzer (Agilent Technologies). An Illumina HiSeq 2500 instrument (Illumina, San Diego, CA, USA) was used for sequencing to produce 51 base reads. Image decomposition and quality values were calculated using the Illumina pipeline module. The miRNAs were identified by aligning the cluster reads with the reference genome using miRBase v. 22.1. NGS sequencing analysis was conducted using Macrogen (Seoul, Republic of Korea).

### 2.8. qPCR Analysis of miRNAs

After adjusting the miRNA concentration to 10 ng, cDNA was synthesized using a TaqMan Advanced miRNA cDNA synthesis kit (Thermo Fisher Scientific, Waltham, MA, USA) following the manufacturer’s instructions. Quantification of miRNAs was performed following the manual for TaqMan Advanced miRNA assays (Thermo Fisher Scientific, Waltham, MA, USA). All analyzed miRNAs were normalized using spiked-in exogenous miRNAs (cel-miR-2-3p from *C. elegans*), which have no sequence similarity to human miRNAs [24,25]. The expression levels of the miRNAs were evaluated using comparative cycle threshold (ΔCt), where ΔCt = Ct ^cel-miR-2-3p^ − Ct ^miRNA^.

### 2.9. Statistical Analysis

Statistical significance of the comparative analyses was determined using Edge-R and Mann–Whitney U-tests (jamovi v. 2.3.18, https://www.jamovi.org/). The comparative analyses were focused on stages I and II NSCLC relative to benign lung-nodule cohorts. Statistical significance was set at *p* < 0.05 and FC = |log_2_ fold change| > 2.0. The diagnostic potential of the miRNA panels identified in the optimization phase was predicted using GraphPad Prism v 10.02 by considering the area under the receiver operating characteristics (ROC) curves (AUC). Identification of the optimum miRNA was helped by the self-developed UDR platform based on a sequential backward search algorithm [26]. An example of a UDR calculation is described in Appendix A.

### 2.10. Bioinformatics

Target prediction for the 15 miRNAs identified in the discovery phase was performed using miRTargetLink 2.0 database [27]. Interaction analysis was conducted in unidirectional mode, initially identifying 3069 target genes. To ensure higher confidence, the results were filtered by applying the “strong evidence” setting and requiring a minimum of two shared targets, yielding a refined set of 20 genes. Network visualization was performed in concentric mode, and node highlighting was applied to emphasize microRNAs and genes with higher connectivity. For functional interpretation, enrichment analysis and annotation were conducted using the miEAA 2.0 tool [28]. We note that the specific algorithms implemented by miRTargetLink 2.0 are not disclosed by the developers.

## 3. Results

### 3.1. Participants’ Clinical Characteristics

The clinical characteristics of the 76-sample cohort in the discovery phase are listed in Table 1. Among the 125 samples initially allotted for NGS analysis, 49 failed the QC test and were discarded, leaving 76 for analysis. The average age of the patients was 73.7 years, with 68.4% being men and 65.8% having a smoking history. Among patients with NSCLC, 46.8% had adenocarcinoma, whereas 51.7% had squamous cell carcinoma. In the benign group, nine patients had benign nodules, including granuloma and aspergilloma, five had pulmonary tuberculosis, and two had pneumonia. In the validation set, the average age of patients was 73.0 years, with 69.3% being men and 68% having a smoking history. Among patients with NSCLC, 55% had adenocarcinoma, whereas 45% had squamous cell carcinoma. In the benign group, 12 patients had benign nodules, including granuloma, fibrosis, and anthracofibrosis; 13 had chronic inflammation, including pulmonary tuberculosis, nontuberculous mycobacterial infection, and sarcoidosis; and 10 had acute infection, including pneumonia, lung abscess, and toxocariasis. In both discovery and validation cohorts, no significant difference was observed in age, sex, and smoking status between the benign group and patients with NSCLC.

### 3.2. Discovery of Differentially Expressed miRNAs Using NGS

miRNA analysis was performed under a strict QC test of the samples. All human-origin miRNAs (*n* = 2656) were registered in miRBase v 22.1 were screened. miRNAs with zero read count >50% in all samples were excluded, leaving 661 mature miRNAs (Appendix A). To enhance data quality, further QC was performed by removing miRNAs with zero read count > 45%, finally leaving 591 miRNAs for statistical analysis. To obtain accurate discrimination for early-stage NSCLC, two analytical methods, Edge-R and Mann–Whitney U test, were applied simultaneously to compare patients with stages I and II NSCLC and those with benign lung nodules (Figure 1). Edge-R analysis (FC = |log_2_ fold change| > 2, *p* < 0.05) resulted in 21 miRNAs that were significantly differentiated in the comparative expression levels of stages I and II NSCLC group compared with the benign group (Table 2). The comparative analyses of each respective stage against benign are presented in Appendix A. Volcano plots of the 591 miRNAs are shown in Appendix A, with the 21 miRNAs in red dots. Additionally, the Mann–Whitney U-test was applied to the same NGS data, resulting in 64 miRNAs with a *p*-value < 0.05 (Appendix A). Fourteen miRNAs were common in the Edge-R and Mann-Whitney U-test analyses (Appendix A). Among the 14 miRNAs, miR-6807-5p and miR-874-5p were excluded, as they did not meet *p* < 0.05 in the comparative analysis of stages I–IV against the benign, leaving 12 miRNAs. Three miRNAs, miR-497-5p, miR-21-5p, and miR-205-5p that have been repeatedly reported in other studies on LC [29,30,31,32,33,34] were added, totaling 15 miRNAs to be examined in the validation phase (Table 3).

### 3.3. Validation of miRNAs Using qPCR

The validity of the 15 miRNAs selected in the discovery phase was studied by qPCR analysis. The details of the sample providers are presented in Table 1. As an exogenous spiked-in control, cel-miR-2-3p from *C. elegans* was applied, which has no sequence similarity to miRNAs of human origin [25]. The comparative analysis of the stages I and II NSCLC group versus the benign group identified six miRNAs, miR-150-5p, miR-1976, miR-301b-3p, miR-369-3p, miR-497-5p, and miR-610 (Mann-Whitney U-test *p* < 0.05 and FC > 1.5) (Figure 2). The FC and *p* values from the comparative analysis of the stages I and II NSCLC group versus the benign group are described in Table 3. The FC values of the downregulated miRNAs are in red. Six miRNAs, miR-150-5p, miR-1976, miR-301b-3p, miR-369-3p, miR-497-5p, and miR-610, were identified to be *p* < 0.05 and FC > 1.5. Except miR-150-5, the rest of the miRNAs were down-regulated (take note that y-axis is in ΔCt). The expression levels and diagnostic potential (AUC) of the six miRNAs are described in Figure 2a,b, respectively.

### 3.4. Optimization for miRNA Panels

Optimal combinations of miRNAs were examined using the six miRNAs identified in the validation phase. To enhance the recognition capacity in the comparative analysis, we devised a simple but novel diagnostic platform called “UDR,” which can be calculated by dividing the average expression level of the up-regulated miRNAs by that of the down-regulated miRNAs (Equation (1)). The UDR values of all serum samples were obtained for all the combinations of the expression levels of the up-regulated and down-regulated miRNAs for the six miRNAs, which were then used to evaluate the AUCs of the miRNA combinations. The AUCs of the six-miRNA panel in the discovery and validation phases are shown in Figure 3a. Two best-optimized miRNA panels were selected that produced high AUCs with the relatively small number of constituting miRNAs (Appendix A): the four-miRNA panel of miR-150-5p, miR-301b-3p, miR-369-3p, and miR-497-5p (Figure 3b) and three-miRNA panel of miR-150-5p, miR-301b-3p, and miR-497-5p (Figure 3c). The relative expression levels of each sample group of the four and three miRNA panels are depicted in Figure 3d,e, respectively. Differential expression was more pronounced between early-stage NSCLC and benign samples than between late or total NSCLC and benign samples, reflecting the miRNA filtering process’ focus on early stages.

### 3.5. Bioinformatics Confirmation of the Identified miRNAs

Target gene analysis of the 15 miRNAs revealed that 20 target genes interacted closely with eight miRNAs: miR-21-5p, miR-497-5p, miR-205-5p, miR-150-5p, miR-202-3p, miR-301b-3p, miR-128-1-5p, and miR-610 (Figure 4). To identify the major regulatory relationships, miRNA enrichment analysis and annotation (miEAA 2.0) were applied to determine the target genes that significantly regulated these miRNAs [28]. The bioinformatics-derived interaction network between the eight highly connected miRNAs and their corresponding target genes, identified through the enrichment analysis, is summarized in Appendix A. In addition, a literature-based validation was performed to examine the association characteristics of the 20 target genes—particularly their relevance to lung cancer (LC) and the identified miRNAs—using Google Scholar search (Table 4) [34,35,36,37,38,39,40,41,42,43,44,45,46,47,48,49,50,51,52,53,54,55,56,57,58]. Among these, 17 genes were strongly associated with LC, whereas ERBB2, SP1, and ZEB1 were related to other tumor types.

## 4. Discussion

This four-phase study identified exosomal miRNA panels with promising diagnostic value for early-stage NSCLC. Using the UDR platform, we highlighted a four-miRNA panel (miR-150-5p, miR-301b-3p, miR-369-3p, and miR-497-5p) that achieved high AUC values in both discovery and validation cohorts. These findings suggest that serum-derived exosomal miRNAs, optimized through a simple ratio-based approach, may serve as non-invasive biomarkers to complement imaging in early lung cancer detection. A notable strength of this study is the use of patients with benign lung nodules as controls, reflecting the real diagnostic challenge of distinguishing malignant from benign nodules on CT. In addition, vesicle-encapsulated (exosomal) miRNAs are more stable in circulation [59] and increasingly recognized as reliable biomarkers [60], which enhanced the robustness of our approach. Together, these features support the clinical relevance and translational potential of our findings.

Pastorino et al. demonstrated in the Multicenter Italian Lung Detection (MILD) and BioMILD trials that plasma-based miRNA signatures can complement LDCT, improve specificity, and provide predictive and prognostic value [61,62]. Unlike these multicenter plasma studies, our analysis was based on serum exosomal miRNAs and applied the novel UDR method, resulting in a distinct non-overlapping panel. Several studies have also proposed circulating miRNA panels for the early detection of NSCLC. A serum-based panel with miR-141 achieved an AUC of ~0.83 in pre-diagnostic samples, but was restricted by modest sample size [63]. A five-miRNA plasma panel (let-7a-5p, miR-1-3p, miR-1291, miR-214-3p, miR-375) demonstrated AUCs > 0.74 in over 1600 subjects, underscoring the value of large multicenter validation [20]. More recently, a nine-miRNA plasma signature validated in multicenter LDCT cohorts yielded AUCs of 0.75–0.78, supporting the integration of miRNAs into imaging-based screening programs [64]. In contrast, the present study analyzed serum-derived vesicle-encapsulated miRNAs, used benign lung nodules as clinically relevant controls, and applied a novel UDR ratio-based framework. This approach resulted in a compact four-miRNA panel with AUCs exceeding 0.93 in both discovery and validation cohorts. These differences highlight not only the diversity across previous reports but also the potential advantages of exosomal serum miRNAs combined with a simple and interpretable optimization strategy for early NSCLC detection. Furthermore, exosomal miRNAs may also have clinical utility in detecting pre-invasive lesions such as adenocarcinoma in situ or minimally invasive adenocarcinoma [65], where conventional imaging or cytology may be limited. Given their stability and accessibility in body fluids, exosomal miRNA assays could be integrated into blood- or urine-based screening workflows for non-invasive early detection. In addition, by improving diagnostic specificity and reducing false-positive findings, exosomal miRNA panels may help mitigate overdiagnosis and overtreatment associated with low-dose CT screening of indolent or slow-growing adenocarcinomas.

The individual miRNAs identified in our panel have also been reported in previous studies, supporting their potential biological and clinical relevance. miR-150-5p shows differential expression in NSCLC and is context-dependent, acting either as an oncogene or tumor suppressor [66,67]. For miR-301b-3p, increased plasma levels have been reported as an early NSCLC biomarker (AUC 0.788) [68], whereas in our exosomal serum data it was downregulated compared with benign controls (AUC 0.677). Interestingly, other studies also noted downregulation of vesicle-derived miR-301b-3p in NSCLC compared with healthy controls [69], suggesting that selective vesicle packaging may explain discrepancies between tumor tissue, plasma, and exosomal measurements. Functionally, miR-301b-3p has been linked to oncogenic activity in NSCLC [70]. miR-369-3p has been less extensively studied, but emerging data implicate it in lung cancer biology. Exosomal miR-369 derived from cancer-associated fibroblasts promotes squamous carcinoma progression via NF1-mediated MAPK signaling [71], and it has also been linked to therapeutic resistance in NSCLC [72,73]. In our dataset, miR-369-3p showed diagnostic potential with an AUC of 0.741, further underscoring its relevance. miR-497-5p is generally downregulated in NSCLC and functions as a tumor suppressor. It directly targets SOX5 and AKT2, thereby inhibiting proliferation and invasion and inducing apoptosis [31,74]. Together, these converging lines of evidence suggest that the selected exosomal miRNAs are not only diagnostically useful but also biologically linked to key oncogenic pathways, including Wnt/β-catenin, RhoGTPase/DLC1, MAPK, and PI3K–AKT signaling. Although we did not experimentally validate these mechanisms in the current study, prior mechanistic literature provides support for the biological plausibility of our diagnostic panel.

A multi-component miRNA panel is preferable for complex such as cancers, where mechanisms are multifactorial and single markers are insufficient [75,76]. To address this, we applied the UDR platform, a simple ratio-based framework that calculates, for each patient sample, the average expression of upregulated miRNAs divided by that of downregulated miRNAs, thereby providing an individual score that highlights differences between disease and control groups. Unlike more complex multivariate modeling strategies (e.g., logistic regression or LASSO), UDR is computationally straightforward, less prone to overfitting in small sample cohorts, and easily interpretable for potential clinical application. Nevertheless, rigorous validation and direct benchmarking against established approaches were beyond the scope of the present study and remain important directions for future research. In our dataset, all four selected miRNAs were consistently detectable above the assay threshold, but we acknowledge that undetectable expression could occur in clinical practice. Because the UDR platform is ratio-based and uses the mean of multiple up- or downregulated miRNAs rather than relying on a single marker, the potential impact of one missing value would be minimized; however, this aspect should be further evaluated in future large-scale studies. Target-gene interaction analysis further supported the biological plausibility of our diagnostic panel. Several of the identified miRNAs directly targeted cancer-relevant genes such as PTEN, VEGFA, and BCL2, and five of them overlapped with the validated set, strengthening their consistency across analyses. These bioinformatics findings suggest that the selected exosomal miRNAs are functionally linked to key oncogenic pathways, thereby reinforcing their potential utility as diagnostic biomarkers. Although serum samples from stage III–IV NSCLC patients were also collected and included in the discovery workflow, the present analysis focused primarily on differentiating early-stage NSCLC from benign nodules, as this represents the most clinically relevant diagnostic context for early detection.

This study has several limitations. First, the sample sizes of both the discovery (*n* = 76) and validation (*n* = 75) cohorts were relatively small, and all samples were collected from a single institution, which may limit statistical power and restrict the generalizability of our findings. Larger multicenter studies with independent external validation cohorts will be required to confirm the robustness and clinical applicability of the proposed miRNA panels. Second, potential confounding factors such as smoking status and co-existing pulmonary diseases may also influence circulating miRNA profiles. Although we summarized their distribution within our cohort, the sample size was too small to allow for reliable subgroup analyses. Furthermore, although previous studies have suggested mechanistic roles for our identified miRNAs in NSCLC, we did not experimentally validate these pathways in the current study, and future functional experiments are needed to substantiate their biological plausibility. In addition, our bioinformatics analysis was limited to target prediction and literature-based associations and did not include functional enrichment analyses to systematically reveal co-regulated biological processes or signaling pathways. Finally, while the UDR platform offers a simple and interpretable diagnostic approach, it was not benchmarked against conventional biomarker discovery methods, and further comparative validation is warranted. Despite these limitations, our study provides important preliminary evidence supporting the potential of exosomal miRNA panels and the UDR platform for early-stage NSCLC detection.

In conclusion, this study demonstrates the potential of serum-derived exosomal miRNA panels, optimized through the UDR platform, as non-invasive biomarkers for early-stage NSCLC detection. The compact four-miRNA panel achieved high diagnostic performance and was supported by bioinformatics evidence linking its components to cancer-related pathways. While further large-scale and multicenter validation is needed, our findings provide preliminary evidence that exosomal miRNAs could complement imaging-based screening and contribute to earlier lung cancer diagnosis.

## Figures and Tables

**Figure 1 diagnostics-15-02735-f001:**
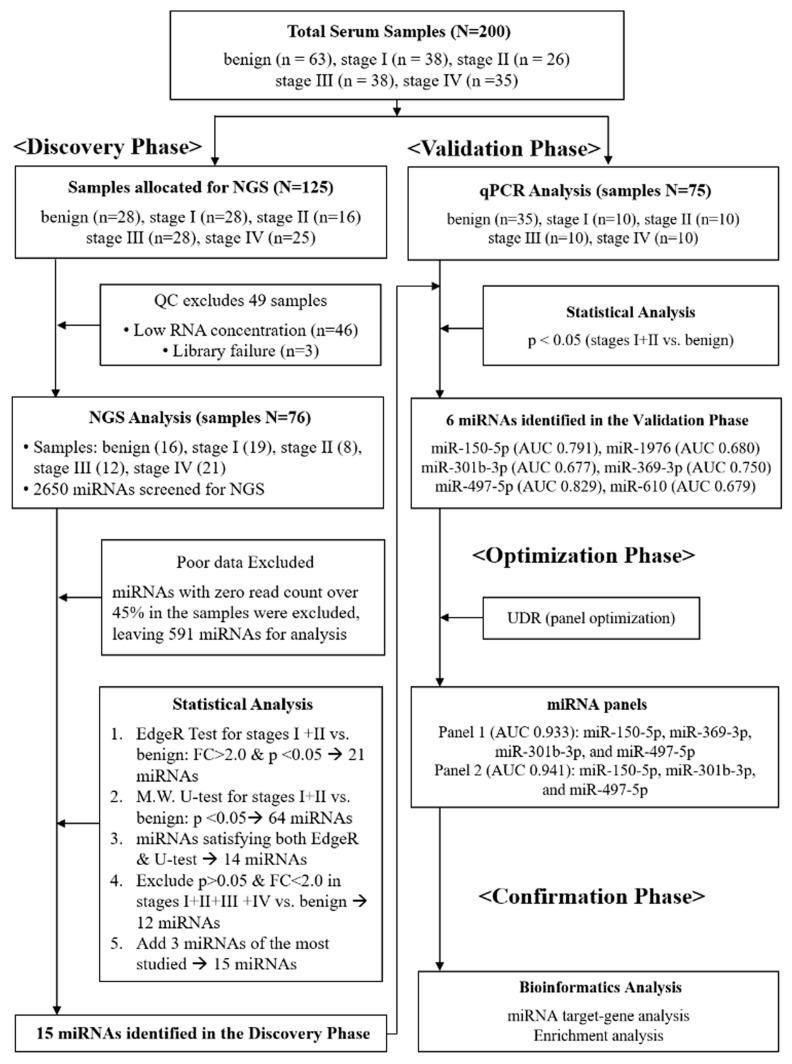
Schematic flow chart describing four-phase procedures applied for this study. In the discovery phase, a total of 125 serum samples were initially allocated for NGS; 49 samples failed quality control (QC) due to low RNA concentration or library failure, leaving 76 samples for analysis. In the validation phase, candidate miRNAs were tested by qPCR in an independent cohort (*n* = 75). In the optimization phase, the up-down ratio (UDR) method was applied to establish diagnostic panels. In the confirmation phase, bioinformatics analysis was conducted to assess diagnostic reliability.

**Figure 2 diagnostics-15-02735-f002:**
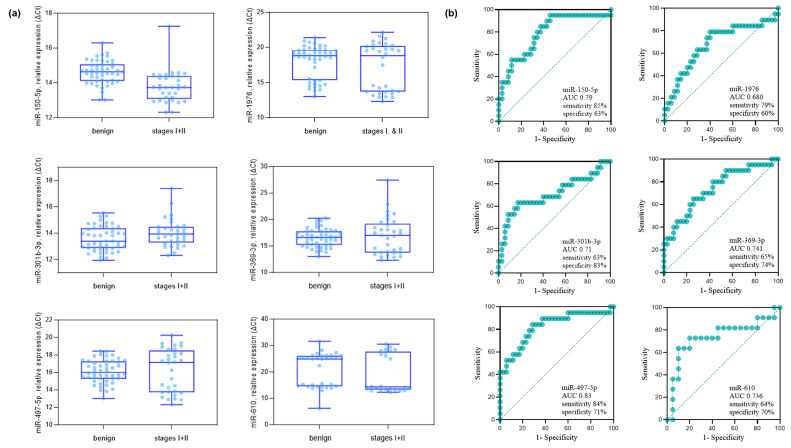
Comparative analysis of benign pulmonary nodules versus stage I–II NSCLC patients in the validation cohort. (**a**) Six miRNAs (miR-150-5p, miR-1976, miR-301b-3p, miR-369-3p, miR-497-5p, and miR-610) were identified by qPCR analysis as significantly different between groups (Mann–Whitney U-test, *p* < 0.05; fold-change > 1.5 based on ΔCt values). The analysis included 35 benign controls and 20 stage I–II NSCLC cases (10 stage I, 10 stage II). (**b**) ROC curves showing the diagnostic performance (AUC, sensitivity, and specificity) of each of the six miRNAs in distinguishing stage I–II NSCLC from benign controls. The AUC (95% CI, *p*-value) values were as follows: miR-150-5p, 0.790 (0.654–0.905, *p* = 0.011); miR-1976, 0.680 (0.515–0.829, *p* = 0.010); miR-301b-3p, 0.710 (0.555–0.865, *p* = 0.010); miR-369-3p, 0.741 (0.548–0.864, *p* = 0.011); miR-497-5p, 0.830 (0.693–0.934, *p* = 0.019); and miR-610, 0.736 (0.612–0.868, *p* = 0.033).

**Figure 3 diagnostics-15-02735-f003:**
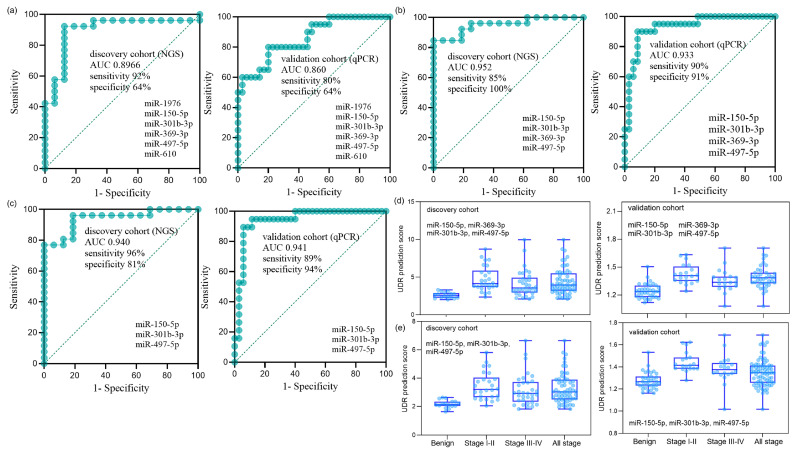
Diagnostic performance of miRNA panels in the discovery and validation cohorts. (**a**) ROC curves of the six-miRNA panel (miR-150-5p, miR-1976, miR-301b-3p, miR-369-3p, miR-497-5p, and miR-610) in the discovery cohort (NGS; *n* = 76: 16 benign, 19 stage I, 8 stage II, 12 stage III, 21 stage IV) and in the validation cohort (qPCR; *n* = 75: 35 benign, 10 stage I, 10 stage II, 10 stage III, 10 stage IV). The AUC (95% CI, *p*-value) was 0.897 (0.776–0.993, *p* < 0.001) for the discovery cohort and 0.860 (0.739–0.948, *p* < 0.001) for the validation cohort. (**b**) ROC curves of the optimized four-miRNA panel (miR-150-5p, miR-301b-3p, miR-369-3p, and miR-497-5p). The AUC was 0.952 (0.872–1.000, *p* < 0.001) for the discovery cohort and 0.933 (0.853–1.000, *p* < 0.001) for the validation cohort. (**c**) ROC curves of the optimized three-miRNA panel (miR-150-5p, miR-301b-3p, and miR-497-5p). The AUC was 0.940 (0.855–0.995, *p* < 0.001) for the discovery cohort and 0.941 (0.863–1.000, *p* < 0.001) for the validation cohort. (**d**,**e**) UDR prediction scores of the four-miRNA panel (**d**) and the three-miRNA panel (**e**), showing distributions across benign controls, stage I+II NSCLC, stage III+IV NSCLC, and all-stage NSCLC in both cohorts. UDR (up-down ratio) was defined as the mean expression level of upregulated miRNAs divided by the mean expression level of downregulated miRNAs (Equation (1)).

**Figure 4 diagnostics-15-02735-f004:**
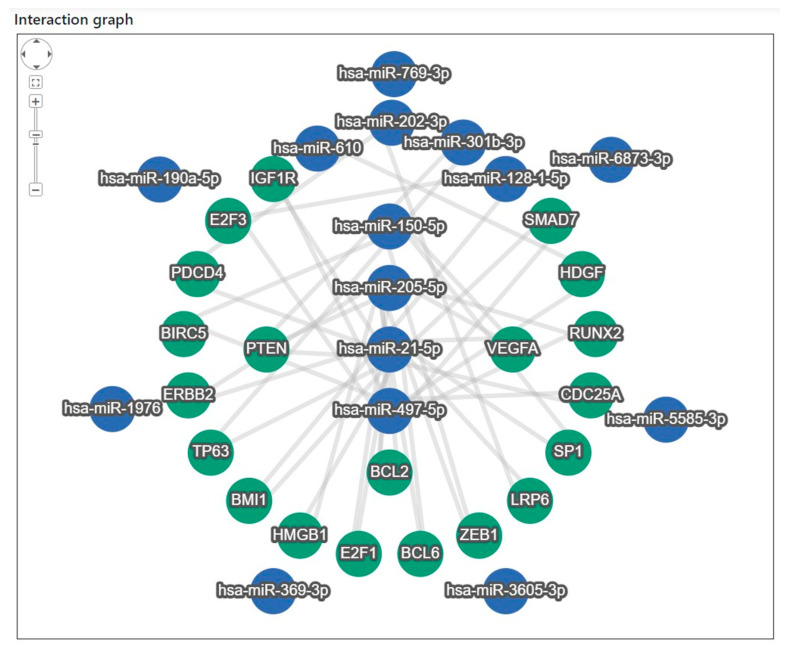
Target gene interaction analysis of the 15 miRNAs identified in the discovery phase. Network analysis revealed 20 target genes interacting with eight miRNAs (miR-21-5p, miR-497-5p, miR-205-5p, miR-150-5p, miR-202-3p, miR-301b-3p, miR-128-1-5p, and miR-610). Target gene enrichment was performed using miEAA 2.0, and associations with lung cancer and other tumor types were examined based on literature searches.

**Table 1 diagnostics-15-02735-t001:** Demographic and clinical characteristics of the patients that participated in this study.

**Discovery Set (*N* = 76)**					
**Variable**	**Number of Patients (%)**	**Whole Group**	**Benign (*N* = 16)**	**NSCLC (*N* = 60)**	** *p* ** **-Value**
Age	mean ± SD	73.7 ± 10.2	72.8 ± 9.7	74.0 ± 10.5	0.680
Sex	Male	52 (68.4)	10 (62.5)	42 (70.0)	0.560
	Female	24 (31.6)	6 (37.5)	18 (30.0)	
Smoking history	No	26 (34.2)	8 (50.0)	18 (30.0)	0.150
	Yes	50 (65.8)	8 (50.0)	42 (70.0)	
Pulmonary comorbidities	No	67 (88.2)	13 (81.3)	54 (90.0)	0.387
	Yes	9 (11.8)	3 (18.8)	6 (10.0)	
Stage	I	N/A	N/A	19 (31.7)	N/A
	II			8 (13.3)	
	III			12 (20.0)	
	IV			21 (35.0)	
Histology	Adeno	N/A	N/A	28 (46.7)	N/A
	SqCC			31 (51.7)	
	NSCLC NOS			1 (1.6)	
**Validation Set (*N* = 75)**					
**Variable**	**Number of Patients (%)**	**Whole Group**	**Benign (*N* = 35)**	**NSCLC (*N* = 40)**	** *p* ** **-Value**
Age	mean ± SD	73.0 ± 11.6	70.5 ± 12.2	75.3 ± 10.8	0.078
Sex	Male	52 (69.3)	23 (65.7)	29 (72.5)	0.618
	Female	23 (30.7)	12 (34.3)	11 (27.5)	
Smoking history	No	24 (32.0)	13 (37.1)	11 (27.5)	0.459
	Yes	51 (68.0)	22 (62.9)	29 (72.5)	
Pulmonary comorbidities	No	67 (89.3)	31 (88.6)	36 (90.0)	0.842
	Yes	8 (10.7)	4 (11.4)	4 (10.0)	
Stage	I	N/A	N/A	10 (25.0)	N/A
	II			10 (25.0)	
	III			10 (25.0)	
	IV			10 (25.0)	
Histology	Adeno	N/A	N/A	22 (55.0)	N/A
	SqCC			18 (45.0)	
	NSCLC NOS			0 (0.0)	

NSCLC, non-small cell lung cancer; SD, standard deviation; N/A, not applicable; Adeno, adenocarcinoma; SqCC, squamous cell carcinoma; NSCLC NOS, non-small cell lung cancer not otherwise specified.

**Table 2 diagnostics-15-02735-t002:** Expression profiles of the miRNAs with significantly differentiated expressions in the discovery phase of the comparative analysis in stages I, II vs. benign. FC values of the down-regulated miRNAs are in red colors.

NGS miRNAs Stages I, II vs. Benign	Edge-R		NGS miRNAs Stages I, II vs. Benign	Edge-R	
	Fold Change	*p* Value		Fold Change	*p* Value
hsa-miR-128-1-5p	3.1	0.0198	hsa-miR-3605-3p	2.6	0.0126
hsa-miR-150-5p	2.0	0.0029	hsa-miR-369-3p	2.6	0.0338
hsa-miR-190a-5p	2.1	0.0123	hsa-miR-4444	3.9	0.0081
hsa-miR-193b-3p	2.4	0.0398	hsa-miR-4732-3p	2.0	0.0366
hsa-miR-1972	2.4	0.0085	hsa-miR-5585-3p	3.0	0.0071
hsa-miR-1976	3.0	0.0004	hsa-miR-610	2.5	0.0234
hsa-miR-202-3p	4.3	0.0010	hsa-miR-6807-5p	2.8	0.0057
hsa-miR-20a-3p	2.4	0.0462	hsa-miR-6873-3p	2.1	0.0431
hsa-miR-2355-5p	2.0	0.0214	hsa-miR-769-3p	2.8	0.0098
hsa-miR-301a-3p	2.1	0.0086	hsa-miR-874-5p	2.0	0.0234
hsa-miR-301b-3p	3.9	0.0001			

**Table 3 diagnostics-15-02735-t003:** Fold change (FC) and *p* values of the 15 miRNAs analyzed by qPCR in the validation phase. FC values of the down-regulated miRNAs are in red colors.

qPCR	Stages I, II vs. Benign		qPCR	Stages I, II vs. Benign	
	FC	*p* Value		FC	*p* Value
hsa-miR-497-5p *	2.5	0.00004	hsa-miR-21-5p *	1.2	0.00015
hsa-miR-369-3p	3.2	0.00266	hsa-miR-150-5p	1.6	0.00027
hsa-miR-301b-3p	1.9	0.01149	hsa-miR-5585-3p	197.7	0.40000
hsa-miR-1976	1.5	0.02746	hsa-miR-202-3p	480.1	0.70000
hsa-miR-610	3.4	0.03188	hsa-miR-205-5p *	1.3	0.80148
hsa-miR-190a-5p	1.5	0.06202	hsa-miR-3605-3p	1.3	1.00000
hsa-miR-769-3p	3.5	0.26385	hsa-miR-128-1-5p	-	-
hsa-miR-6873-3p	1.3	0.43047			

* miRNAs repeatedly studied in other groups. - undetermined data in qPCR analysis.

**Table 4 diagnostics-15-02735-t004:** Target genes identified through literature-based validation of the 15 miRNAs, along with their functions and associated miRNAs.

Target Genes	Functions	Associated microRNA	References
BCL2 and PTEN	Cellular proliferation, invasion, migration, and apoptosis in lung squamous carcinoma are regulated by miR-21 via targeting PTEN, RECK and Bcl-2.	miR-21-5p	[35]
BCL2	Reviews prognostic and predictive effect of Bcl-2 family in NSCLC.	·	[36]
PTEN	Cell migration and invasion of NSCLC cells is inhibited by lncRNA MEG3 by regulating miR-21-5p/PTEN axis.	miR-21-5p	[37]
VEGFA	miR-497 binds to 3′-UTR of VEGFA mRNA in NSCLC to suppress translation.	miR-497-5p	[38]
VEGFA	VEGFA mediates angiogenesis and contributes to cancer growth and metastasis targeting tumor cells including LC.	·	[39]
BCL6	BET inhibition upregulates BCL6 in KRAS-mutant cancers, including NSCLC.	·	[40]
BIRC5	BIRC5 is substantially overexpressed in lung adenocarcinoma.	·	[41]
BMI1	Overexpression of BMI1 is associated with tumor size, poor differentiation, distant metastasis and poor overall survival in NSCLC.	·	[42]
BMI1	miR-128 lowers β-catenin and intracellular signaling pathway-related factors in cancer stem cells, and decreases BMI1.	miR-128	[43]
E2F1	miR-205-5p targets E2F1 to impair SKP2-mediated Beclin1 ubiquitination, promote autophagy and inhibit pulmonary fibrosis in silicosis	miR-205-5p	[44]
E2F3	miR-497 and miR-34a cooperatively facilitate growth of LC cells via downregulation of E2F3.	miR-497-5p	[45]
E2F3	E2F3 overexpression upregulates anti-apoptotic factor, Bcl-2, contributing to uncontrolled tumor growth in NSCLC.	·	[46]
HDGF	Down regulation of miR-497-5p overexpresses HDGF, which is associated with NSCLC tumors and cell lines.	miR-497-5p	[34]
HMGB1	HMGB1 is upregulated in LC by activating Wnt/β-catenin pathway.	·	[47]
IGF1R	In NSCLC, the expression of IGF1R is regulated by MLETA1 and cell mobility is promoted by sponging miR-497-5p.	miR-497-5p	[48]
LRP6	RASSF 10, which is under-expressed in LC, binds to LRP 6 through the coiled-coil domains.	·	[49]
PDCD4 and PTEN	miR-21 mitigates lung injury by reducing PTEN/Foxo-1-TLR4/NF-KB signaling cascade.	miR-21-5p	[50]
RUNX2	Overexpression of RUNX2/p57 in NSCLC and metastatic LC is associated with H3K27Ac at PI gene promoter region.	·	[51]
SMAD7	Carboplatin suppresses miR-21 to inhibit TGFβ receptor signaling mediated NSCLC cell invasion resulting in upregulation of SMAD7.	miR-21-5p	[52]
TP63	Transactivation domain containing p63 is a target gene of miR-301b, which suggests miR-301b may target TAp63 in NSCLC as a oncomir.	miR-301b	[53]
CDC25A	miR-21 deregulates miR-184 and associated with tumor malignancy in NSCLC.	miR-21-5p	[54]
ERBB2 * (HER2)	Stat3 leads to miR-21 expression and breast cancer metastasis via ERBB2 regulation.	miR-21-5p	[55]
ERBB2 * (HER2)	miR-205-5p is significantly downregulated in breast tumors by targeting ERBB3 and ZEB1 oncogenes.	miR-205-5p	[56]
SP1 *	SP1-induced ZFAS1 binds to miR-150-5p, which suppresses colorectal cancer by targeting VEGFA, and causes colorectal cancer progression by upregulating VEGFA.	miR-150-5p	[57]
ZEB1 *	ZEB1 is a master regulator of the EMT phenotype in cancer progression. In osteosarcoma cells, MIAT may induce EMT via miR-150/ZEB1 pathway.	miR-150-5p	[58]

* Genes associated with tumors other than lung cancer.

## Data Availability

The data supporting the findings of this study are available upon reasonable request from the corresponding author. The data are not publicly available due to patient confidentiality and institutional restrictions.

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
