# Peer review of "Exosomal microRNA Panels for Detecting Early-Stage Non-Small Cell Lung Cancer"

_diagnostics, 2025, doi:10.3390/diagnostics15212735_

Round 1
Reviewer 1 Report (Previous Reviewer 2)
Comments and Suggestions for Authors
In this resubmitted version, the authors revised the figure legends and provided in depth discussions on the study strategy, results and limitations, making the manuscript now much clearer. The authors have addressed all of my previous concerns.
Author Response
Reviewer #1
In this resubmitted version, the authors revised the figure legends and provided in depth discussions on the study strategy, results and limitations, making the manuscript now much clearer. The authors have addressed all of my previous concerns.
Response: We sincerely thank the reviewer for the positive evaluation of our revised manuscript. We are pleased that the additional clarifications and revisions to the figures, discussion, and limitations have resolved the reviewer’s previous concerns.
Reviewer 2 Report (New Reviewer)
Comments and Suggestions for Authors
This study aims to screen serum exosomal miRNA panels for the non-invasive diagnosis of early-stage non-small cell lung cancer (NSCLC). The research topic holds clear clinical value. However, the manuscript presents serious issues in logic, methodological rigor, data authenticity, and consistency, which significantly undermine the credibility and reproducibility of the findings.
① Which specific stage does "early-stage lung cancer" refer to? Early-stage lung cancers often have relatively limited blood supply. How was it ensured that sufficient exosomes were present in the blood for detection? Is there any related evidence supporting this?
② How were the "patients with benign pulmonary nodules" diagnosed? Was their benign nature definitively confirmed? Why was a control group of healthy individuals not included?
③ There are multiple inconsistencies in the descriptions throughout the manuscript, causing confusion for the reader. For example: Section 3.1 in the text states that "among the 125 samples initially allocated for NGS analysis, 49 failed QC... leaving 76 for analysis," yet Figure 1's flowchart indicates "Discovery Phase NGS, sample QC (N=125)" and "Validation Phase qPCR Analysis (samples N=75)." This easily leads to the misunderstanding that 125 samples were used for NGS.
④ The Discussion section includes extensive literature reviews that are not closely related to the core findings of this study, which distracts from the main focus of the article.
⑤ The bioinformatics analysis remains superficial, limited to prediction and literature associations. It lacks functional enrichment analysis and fails to systematically reveal the biological processes or signaling pathways that these four miRNAs may co-regulate.
⑥ It is recommended to provide clearer images.
⑦The study also collected and tested serum from stage III and IV lung cancer patients. Why were the results related to these stages not presented or discussed in the findings? Their inclusion in the methodology raises questions about the analytical strategy and the completeness of the reported results. Clarifying the purpose and fate of this data is crucial.
Author Response
Reviewer #2
This study aims to screen serum exosomal miRNA panels for the non-invasive diagnosis of early-stage non-small cell lung cancer (NSCLC). The research topic holds clear clinical value. However, the manuscript presents serious issues in logic, methodological rigor, data authenticity, and consistency, which significantly undermine the credibility and reproducibility of the findings.
Response: We sincerely thank the reviewer for this overall assessment. We acknowledge these important concerns and have carefully revised the manuscript to address each issue in detail. Specifically, we have clarified our definition of “early-stage” NSCLC, provided further justification and references for exosomal miRNA detection, explained the diagnosis of benign nodules and rationale for control selection, corrected inconsistencies in sample descriptions and figures, streamlined the Discussion to improve focus, expanded the bioinformatics analysis description, improved figure quality, and clarified the role of stage III–IV samples. Detailed responses to each specific comment are provided below.
① Which specific stage does "early-stage lung cancer" refer to? Early-stage lung cancers often have relatively limited blood supply. How was it ensured that sufficient exosomes were present in the blood for detection? Is there any related evidence supporting this?
Response: We defined “early-stage NSCLC” as stage I–II disease, consistent with widely accepted classifications. This has now been clarified in the Methods section. While it is true that early tumors may have relatively limited vascularization, previous studies have demonstrated that tumor-derived exosomes are abundantly secreted into the bloodstream even in early-stage cancer, and circulating exosomal miRNAs can be reliably detected [1,2]. For example, exosomal miRNA signatures have been successfully identified in stage I lung cancer patients [3]. These findings support the feasibility of detecting exosomal miRNAs in our study population.
② How were the "patients with benign pulmonary nodules" diagnosed? Was their benign nature definitively confirmed? Why was a control group of healthy individuals not included?
Response: We thank the reviewer for this insightful comment. The “benign pulmonary nodule” group in our study included patients whose benign nature was definitively confirmed through histopathological examination following biopsy or surgical resection. These patients typically underwent tissue confirmation because their imaging findings were indeterminate and malignancy could not be excluded at presentation. Therefore, this group represents the key clinical population in which non-invasive biomarkers are most needed - patients presenting with small or ambiguous pulmonary nodules that require differentiation between malignant and benign lesions. We intentionally did not include healthy individuals as controls because they rarely undergo chest CT or nodule evaluation in clinical practice, and comparisons against healthy subjects would not reflect the real-world diagnostic challenge faced in early lung cancer detection. Using benign nodule patients as controls thus enhances the clinical relevance and translational applicability of our findings. This clarification has now been added to the Methods section to specify that all benign pulmonary nodules were pathologically confirmed.
③ There are multiple inconsistencies in the descriptions throughout the manuscript, causing confusion for the reader. For example: Section 3.1 in the text states that "among the 125 samples initially allocated for NGS analysis, 49 failed QC... leaving 76 for analysis," yet Figure 1's flowchart indicates "Discovery Phase NGS, sample QC (N=125)" and "Validation Phase qPCR Analysis (samples N=75)." This easily leads to the misunderstanding that 125 samples were used for NGS.
Response: We thank the reviewer for pointing out this potential source of confusion. We have revised Figure 1 to clarify the workflow. Specifically, the box in the discovery phase now indicates “Samples allocated for NGS (N=125)”, followed by a separate box specifying “QC Excluded (N=49)”, which results in “NGS analysis (N=76)”. This modification ensures consistency with the description in Section 3.1 and eliminates any misunderstanding that all 125 samples underwent NGS. We believe this revision makes the figure clearer and more consistent with the text (Revised Figure 1).
④ The Discussion section includes extensive literature reviews that are not closely related to the core findings of this study, which distracts from the main focus of the article.
Response: We thank the reviewer for this constructive comment. We have carefully revised the Discussion to improve focus on the core findings. Specifically, we removed less relevant references and shortened descriptive passages to avoid unnecessary detail. The revised section now emphasizes studies most directly comparable to our work and integrates them more concisely, while still providing sufficient context for readers. These changes streamline the narrative and reduce redundancy, ensuring that the Discussion remains closely aligned with the central results of our study.
⑤ The bioinformatics analysis remains superficial, limited to prediction and literature associations. It lacks functional enrichment analysis and fails to systematically reveal the biological processes or signaling pathways that these four miRNAs may co-regulate.
Response: We thank the reviewer for this important comment. We agree that our bioinformatics analysis was limited to target prediction and literature-based associations, and did not include detailed functional enrichment. We have now explicitly acknowledged this limitation in the revised Discussion, emphasizing that future studies should incorporate systematic enrichment and experimental validation to better delineate the biological processes co-regulated by the identified miRNAs.
⑥ It is recommended to provide clearer images.
Response: We thank the reviewer for this helpful suggestion. All figures have been replaced with higher-resolution versions in the revised manuscript to improve clarity and readability.
⑦ The study also collected and tested serum from stage III and IV lung cancer patients. Why were the results related to these stages not presented or discussed in the findings? Their inclusion in the methodology raises questions about the analytical strategy and the completeness of the reported results. Clarifying the purpose and fate of this data is crucial.
Response: We thank the reviewer for this important comment. In both the discovery and validation cohorts, serum samples were collected from patients with stage I–IV NSCLC as well as benign pulmonary nodules. This design was intentional, since at the time of collection it was uncertain which stage-specific patterns would emerge, and we aimed to build a comprehensive biobank for future analyses. However, the central objective of the present study was to establish diagnostic panels for early-stage NSCLC (stage I–II) versus benign nodules, reflecting the clinical scenario where distinguishing malignant from benign nodules is most critical. For this reason, the results presented here focus on stage I–II disease. The stage III–IV samples were included in the workflow and contributed to candidate identification in the discovery phase, but detailed results for advanced stages were not reported, as they were outside the scope of this manuscript. We have clarified this rationale in the revised Discussion section.
References
- Sui, S.; Xu, C.; Kanda, M.; Okugawa, Y.; Toiyama, Y.; Park, J.O.; Hur, H.; Kim, S.C.; Taketomi, A.; Kodera, Y. Exosomal liquid biopsy for the early detection of gastric cancer: the DESTINEX multicenter study. JAMA surgery 2025, 160, 973-982.
- Zhang, Z.; Lin, F.; Wu, W.; Jiang, J.; Zhang, C.; Qin, D.; Xu, Z. Exosomal microRNAs in lung cancer: a narrative review. Translational Cancer Research 2024, 13, 3090.
- Jin, X.; Chen, Y.; Chen, H.; Fei, S.; Chen, D.; Cai, X.; Liu, L.; Lin, B.; Su, H.; Zhao, L. Evaluation of tumor-derived exosomal miRNA as potential diagnostic biomarkers for early-stage non–small cell lung cancer using next-generation sequencing. Clinical cancer research 2017, 23, 5311-5319.
Reviewer 3 Report (New Reviewer)
Comments and Suggestions for Authors
Kim et al have undertaken a study to investigate potential blood-based biomarkers of early stage NSCLC. The search for such biomarkers is clinically important as it could prevent patients from needing an unnecessary biopsy. The authors have taken the approach of focussing on miRNAs encapsulated in extracellular vesicles as these could be more stable for analysis. Using discovery and validation cohorts, along with a novel approach to combine biomarkers, they have proposed a 3 and a 4 marker combination with promise in NSCLC.
The version of the manuscript I received had tracked changes on it, suggesting that modifications had already been made to several sections. As a result, I just have some minor points to raise:
- The IRB number is missing from methods section 2.2.
- Methods section 2.4 describing exosome isolation states that precipitation buffer is added to the pellet, but according to the manufacturers, the precipitation buffer is added to the cell-free serum (the pellet is discarded). This should be checked and corrected.
- Similarly in Methods section 2.5 describing RNA extraction – was QIAzol added to serum as stated, or was it added to the resuspended isolated EVs?
- Currently, Figure 1 is first listed in the Materials and Methods (2.1), but not shown until the results under section 3.2. It would make more sense for the reader for this flow chart to be shown earlier. Where it is currently placed is too late as it is after the description of the discovery results.
- Table 4 summarises target genes for the 15 miRNAs identified in the discovery part of the study. However, some of the target genes listed e.g. RUNX2 and BCL6 have no associated miRNA listed in the table. They do however have miRNAs listed in Supplementary Table 6. Why is there a discrepancy between these two tables?
- Supplementary data – the text currently states under the Rationale for selecting the two best miRNA panels: “the third-ranked panel is from the three-miRNA panel of miR-150, miR-301b-3p, and miR-497-5p with only negligibly minute difference of 0.0004.” However, the third-ranked panel is from a 4-miRNA panel. I think the statement above has been mistyped as it mentions 4 miRNAs later, but it should be looked at and corrected.
Author Response
Reviewer #3
Kim et al have undertaken a study to investigate potential blood-based biomarkers of early stage NSCLC. The search for such biomarkers is clinically important as it could prevent patients from needing an unnecessary biopsy. The authors have taken the approach of focussing on miRNAs encapsulated in extracellular vesicles as these could be more stable for analysis. Using discovery and validation cohorts, along with a novel approach to combine biomarkers, they have proposed a 3 and a 4 marker combination with promise in NSCLC.
The version of the manuscript I received had tracked changes on it, suggesting that modifications had already been made to several sections. As a result, I just have some minor points to raise:
Response: We thank the reviewer for the positive assessment and constructive summary of our study. We appreciate the acknowledgment of the clinical relevance and methodological strengths of our approach. All tracked changes and revisions noted in the manuscript were made in response to the previous review round, and the current version incorporates additional clarifications and formatting refinements as requested by the reviewer and the editor. Detailed responses to the minor points raised by the reviewer are provided below.
1. The IRB number is missing from methods section 2.2.
Response: We thank the reviewer for pointing this out. The Institutional Review Board (IRB) approval number has now been added to Section 2.2 of the Methods.
2. Methods section 2.4 describing exosome isolation states that precipitation buffer is added to the pellet, but according to the manufacturers, the precipitation buffer is added to the cell-free serum (the pellet is discarded). This should be checked and corrected.
Response: We thank the reviewer for pointing out this important detail. The description in the Methods section has been corrected accordingly. In the revised version, we now specify that the precipitation buffer was added to the cell-free serum, not to the pellet, in accordance with the manufacturer’s protocol. The updated text reads as follows:
“Briefly, 1 mL of serum sample was centrifuged at 3,000 × g for 10 min to obtain cell-free serum. Precipitation Buffer A (200 μL) was gently mixed with the cell-free serum and incubated for 1 h at 4 °C. Exosomes were pelleted by centrifugation at 1,500 × g for 30 min at 20 °C, and the resulting pellet was resuspended in 270 μL of Resuspension Buffer by vortexing.”
This revision ensures that the procedure accurately reflects the manufacturer’s instructions and corrects the previous misstatement.
3. Similarly in Methods section 2.5 describing RNA extraction – was QIAzol added to serum as stated, or was it added to the resuspended isolated EVs?
Response: We appreciate the reviewer’s careful observation. As correctly noted, QIAzol was added to the resuspended exosome (EV) suspension, not directly to serum. The description in the Methods section has been revised to reflect this correction.
4. Currently, Figure 1 is first listed in the Materials and Methods (2.1), but not shown until the results under section 3.2. It would make more sense for the reader for this flow chart to be shown earlier. Where it is currently placed is too late as it is after the description of the discovery results.
Response: As recommended, Figure 1 has been repositioned immediately after Section 2.1 (Study Design) to allow readers to visualize the overall workflow earlier in the manuscript.
5. Table 4 summarises target genes for the 15 miRNAs identified in the discovery part of the study. However, some of the target genes listed e.g. RUNX2 and BCL6 have no associated miRNA listed in the table. They do however have miRNAs listed in Supplementary Table 6. Why is there a discrepancy between these two tables?
Response: We thank the reviewer for this thoughtful comment. The apparent discrepancy between Table 4 and Supplementary Table 6 arises from the difference in analytic approaches used for these datasets. Table 4 presents the results of a literature-based validation of the 20 target genes identified from the 15 miRNAs. Each gene’s relevance to lung cancer (LC) and its association with the identified miRNAs were confirmed through a Google Scholar search. Seventeen genes were found to be strongly associated with LC, whereas three genes (ERBB2, SP1, and ZEB1) were related to other tumor types. The title of Table 4 has been revised to clearly indicate its literature-based nature (“Target genes identified through literature-based validation of the 15 miRNAs, along with their functions and associated miRNAs”). Supplementary Table 6, on the other hand, summarizes the bioinformatics-derived interaction network between the eight highly connected miRNAs (miR-21-5p, miR-497-5p, miR-205-5p, miR-150-5p, miR-202-3p, miR-301b-3p, miR-128-1-5p, and miR-610) and their corresponding target genes, as obtained from the miRNA enrichment analysis (miEAA 2.0). Accordingly, Section 3.5 (“Bioinformatics confirmation of the identified miRNAs”) have been revised to clarify that Supplementary Table 6 represents bioinformatics-based network results, whereas Table 4 summarizes literature-validated associations.
6. Supplementary data – the text currently states under the Rationale for selecting the two best miRNA panels: “the third-ranked panel is from the three-miRNA panel of miR-150, miR-301b-3p, and miR-497-5p with only negligibly minute difference of 0.0004.” However, the third-ranked panel is from a 4-miRNA panel. I think the statement above has been mistyped as it mentions 4 miRNAs later, but it should be looked at and corrected.
Response: We appreciate the reviewer’s careful observation. The statement indeed contained a typographical error. It has been corrected in the revised Supplementary Materials.
This manuscript is a resubmission of an earlier submission. The following is a list of the peer review reports and author responses from that submission.
Round 1
Reviewer 1 Report
Comments and Suggestions for Authors
The present manuscript, entitled "Exosomal microRNA panels for detecting early-stage non-small cell lung cancer" by Young Jun Kim et al., addresses the significant clinical requirement for minimally invasive biomarkers to detect early-stage NSCLC. The utilisation of a four-phase study (discovery, validation, optimisation, and confirmation) and the combination of NGS and qPCR for miRNA profiling are commendable. Furthermore, the implementation of a novel "UDR" platform for panel optimisation demonstrates an innovative approach.
Nevertheless, the manuscript exhibits numerous substantial deficiencies, thereby compromising the reliability and the significance of its conclusions. Firstly, the absence of detailed descriptions of patient selection criteria, inclusion/exclusion criteria, and randomisation methods limits the transparency of the study and raises questions about the validity of the patient cohorts. The sample sizes in both the discovery (n=76) and validation (n=75) phases are small and likely to be underpowered for the purpose of discovering biomarkers in such a heterogeneous disease, which in turn diminishes the robustness of the reported findings.
A significant concern pertains to the absence of an independent external validation cohort. The restriction of the sample to a single institution has a considerable impact on the generalizability of the conclusions. While the authors report high AUC values (over 0.93) for their panels of microRNA, they do not provide confidence intervals or undertake direct statistical comparisons with existing panels, such as those from the MILD trial. Furthermore, the study does not adequately address confounding factors, such as smoking status or co-existing pulmonary diseases, which are known to influence the profile of microRNAs.
The discussion section does not provide a thorough comparison to existing literature and often makes claims that are not fully supported by the data. The potential biological mechanisms by which the identified miRNAs may contribute to the pathogenesis of NSCLC remain to be elucidated, thus impeding the biological plausibility of the panel. Furthermore, there is an absence of empirical experimentation or mechanistic insights that could serve to substantiate the proposed biomarkers. Moreover, this section is deficient in terms of the study's limitations. From a technical perspective, while the figures and tables are generally clear, some of the legends lack context, which hinders the interpretation of the data independently. Minor linguistic issues are present, though these do not detract significantly from the clarity of the text. The UDR platform, while conceptually interesting, is presented without rigorous validation or benchmarking against established methods. The text makes no direct comparisons to traditional biomarker discovery approaches, and the justification for the UDR method's superiority remains unconvincing. The bioinformatics analyses, including target prediction and interaction networks, are described only superficially. The absence of critical details regarding databases, thresholds, and algorithms used has a detrimental effect on the interpretability and reproducibility of these results.
In conclusion, although the manuscript addresses a clinically relevant topic and applies some innovative methodologies, it lacks sufficient methodological rigor, external validation, and in-depth biological exploration to support the robustness and generalizability of its findings. Future revisions should include more comprehensive patient cohort descriptions, independent validation in larger and more diverse populations, and deeper bioinformatics and functional analyses of the identified microRNAs. The study under review here constitutes an interesting preliminary exploration of the subject, but it does not yet reach the level of evidence required for a clinically actionable biomarker.
Comments on the Quality of English Language
On occasion, the language used is not always as precise as it could be, with occasional instances of grammatical inaccuracy.
Author Response
Reviewer #1
- The present manuscript, entitled "Exosomal microRNA panels for detecting early-stage non-small cell lung cancer" by Young Jun Kim et al., addresses the significant clinical requirement for minimally invasive biomarkers to detect early-stage NSCLC. The utilisation of a four-phase study (discovery, validation, optimisation, and confirmation) and the combination of NGS and qPCR for miRNA profiling are commendable. Furthermore, the implementation of a novel "UDR" platform for panel optimisation demonstrates an innovative approach.
Response: We sincerely thank the reviewer for these encouraging comments. We are pleased that the overall study design and methodology, including the four-phase approach and the integration of NGS and qPCR with the UDR optimization platform, were well appreciated. These strengths, we believe, support the novelty and translational value of our work, and we are grateful that the reviewer has recognized them.
- Nevertheless, the manuscript exhibits numerous substantial deficiencies, thereby compromising the reliability and the significance of its conclusions. Firstly, the absence of detailed descriptions of patient selection criteria, inclusion/exclusion criteria, and randomisation methods limits the transparency of the study and raises questions about the validity of the patient cohorts.
Response: We sincerely thank the reviewer for this important comment. We agree that a detailed description of patient selection is essential for transparency and reproducibility. To address this concern, we have substantially revised the Methods section. Specifically, we now clearly describe the discovery and validation cohorts, including the enrollment periods, sample sizes, and disease categories. We have also added explicit inclusion and exclusion criteria. Patients were included if they had histologically confirmed stage I–IV NSCLC or benign pulmonary disease and provided written informed consent. Patients were excluded if their diagnosis or staging was uncertain, if clinical information was insufficient, or if serum sample quality was inadequate. All samples were collected at diagnosis before initiation of any treatment. With respect to randomization, we have clarified that this study was based on prospectively collected clinical samples, and that cohorts were defined by the time of collection rather than by random assignment. To avoid confusion, we have removed the previous statement referring to “random assignment” from the Methods. These revisions provide a more transparent and comprehensive description of patient selection, thereby improving the clarity and validity of the study design.
- The sample sizes in both the discovery (n=76) and validation (n=75) phases are small and likely to be underpowered for the purpose of discovering biomarkers in such a heterogeneous disease, which in turn diminishes the robustness of the reported findings. A significant concern pertains to the absence of an independent external validation cohort. The restriction of the sample to a single institution has a considerable impact on the generalizability of the conclusions.
Response: We thank the reviewer for these constructive and important comments. We fully acknowledge that the relatively small sample sizes in the discovery and validation phases may limit statistical power and robustness, and that the lack of an independent external validation cohort from multiple institutions restricts the generalizability of our findings. While additional large-scale or multicenter validation could not be conducted within the scope of this revision, we agree these are critical limitations that should be clearly stated. To address the reviewer’s concern, we have now added a dedicated paragraph on limitations in the Discussion section, explicitly noting the constraints of sample size, single-institution setting, and absence of external validation. Despite these limitations, we believe that our four-phase design—comprising discovery, validation, optimization, and confirmation phases—and the application of the novel UDR platform lend methodological strength and provide important preliminary evidence for the potential utility of exosomal miRNA panels in early-stage NSCLC detection. We emphasize, however, that future large-scale, multicenter studies will be essential to further validate and generalize our findings.
- While the authors report high AUC values (over 0.93) for their panels of microRNA, they do not provide confidence intervals or undertake direct statistical comparisons with existing panels, such as those from the MILD trial.
Response: We thank the reviewer for raising this important point. In the revised manuscript, we have added 95% confidence intervals and p-values for all ROC analyses, which are now presented in the figure legends (Figures 2 and 3). For example, the optimized four-miRNA panel achieved an AUC of 0.952 (95% CI: 0.872–1.000, p < 0.001) in the discovery cohort and 0.933 (95% CI: 0.853–1.000, p < 0.001) in the validation cohort. The optimized three-miRNA panel showed similar performance with an AUC of 0.940 (95% CI: 0.855–0.995, p < 0.001) and 0.941 (95% CI: 0.863–1.000, p < 0.001), respectively. Regarding comparisons with existing panels, we have expanded the Discussion to include contextual analysis of previously reported circulating miRNA signatures, such as those from the MILD and BioMILD trials, as well as more recent plasma-based multicenter studies. While our panel composition differs from these studies—likely reflecting differences in biospecimen source (serum exosomal vs plasma), patient populations, and analytic strategies—the comparable diagnostic performance across independent reports underscores the potential of circulating miRNAs as non-invasive biomarkers to complement LDCT for early NSCLC detection.
- Furthermore, the study does not adequately address confounding factors, such as smoking status or co-existing pulmonary diseases, which are known to influence the profile of microRNAs.
Response: We appreciate the reviewer’s insightful comment. We agree that smoking status and co-existing pulmonary diseases are potential confounders that could affect circulating miRNA profiles. We have therefore reviewed and summarized the distribution of smoking history and major pulmonary comorbidities in our cohort. The distributions were generally comparable between NSCLC patients and benign nodule controls. However, given the limited sample size, detailed subgroup analyses stratified by smoking or comorbidity status were not feasible, and we have acknowledged this limitation in the Discussion section. We believe that future large-scale studies will be needed to fully address the potential impact of these confounders.
- The discussion section does not provide a thorough comparison to existing literature and often makes claims that are not fully supported by the data.
Response: We thank the reviewer for this valuable comment. We agree that a more detailed comparison with existing literature would strengthen the Discussion. In the revised manuscript, we have substantially expanded the Discussion to incorporate several additional recent studies. A plasma-based 24-miRNA panel showed a corrected AUC of 0.78, although the large panel size may limit clinical applicability [1]. A smaller serum-based panel including miR-141 achieved an AUC of ~0.83 in pre-diagnostic samples, but was restricted by modest sample size [2]. Genome-wide profiling in more than 3,000 individuals reached 91.4% diagnostic accuracy, yet required complex sequencing and computational modeling [3]. A five-miRNA plasma panel (let-7a-5p, miR-1-3p, miR-1291, miR-214-3p, miR-375) demonstrated AUCs >0.74 in over 1,600 subjects, underscoring the value of large multicenter validation [4]. Subtype-specific circulating miRNA panels derived from more than 4,000 patients emphasized the heterogeneity of NSCLC subtypes [5]. More recently, a nine-miRNA plasma signature validated in multicenter LDCT cohorts yielded AUCs of 0.75–0.78, supporting the integration of miRNAs into imaging-based screening programs [6]. We also explicitly highlight how our study differs from these prior reports in terms of sample source (serum exosomal vs. plasma), control group (benign lung nodules vs. healthy individuals), and analytic framework (ratio-based UDR vs. multivariate modeling). Furthermore, we carefully revised passages that might have overstated the robustness of our results, and now emphasize that our findings should be considered preliminary until validated in larger, independent cohorts. These revisions strengthen the contextualization of our findings, demonstrate awareness of the rapidly evolving literature, and ensure that our claims are consistent with the data presented.
- The potential biological mechanisms by which the identified miRNAs may contribute to the pathogenesis of NSCLC remain to be elucidated, thus impeding the biological plausibility of the panel. Furthermore, there is an absence of empirical experimentation or mechanistic insights that could serve to substantiate the proposed biomarkers.
Response: We thank the reviewer for this important comment. We agree that elucidating the biological mechanisms underlying the identified miRNAs would strengthen the plausibility of our diagnostic panel. While the present study was designed as a clinical biomarker discovery and validation study rather than a mechanistic investigation, we have revised the Discussion to incorporate relevant evidence from prior literature that supports the biological plausibility of our findings. Specifically, prior studies have shown that miR-150-5p regulates genes such as SRCIN1, HMGA2, and β-catenin, influencing proliferation, metastasis, and recurrence; miR-301b-3p targets the tumor suppressor DLC1 to promote proliferation and invasion; miR-369-3p participates in cancer-associated fibroblast–mediated MAPK activation and contributes to drug resistance through targets such as SLC35F5 and PTPN12; and miR-497-5p functions as a tumor suppressor by targeting SOX5, AKT2, and BCL2, thereby regulating apoptosis and proliferation. These mechanistic links provide biological plausibility for the involvement of the identified miRNAs in NSCLC pathogenesis. We also acknowledge that our study did not include experimental validation of these pathways, and we have now added this point explicitly to the Limitations section of the Discussion.
- Moreover, this section is deficient in terms of the study's limitations. From a technical perspective, while the figures and tables are generally clear, some of the legends lack context, which hinders the interpretation of the data independently.
Response: We thank the reviewer for these comments. We have substantially revised the Limitations section of the Discussion to address the reviewer’s concerns. In addition, we have revised all figure and table legends to include sufficient details, so that each can be interpreted independently.
- Minor linguistic issues are present, though these do not detract significantly from the clarity of the text. The UDR platform, while conceptually interesting, is presented without rigorous validation or benchmarking against established methods. The text makes no direct comparisons to traditional biomarker discovery approaches, and the justification for the UDR method's superiority remains unconvincing.
Response: We thank the reviewer for this valuable comment. We have carefully polished the language throughout the manuscript to improve clarity and fluency. Regarding the UDR platform, we acknowledge that our study did not include direct benchmarking against other established biomarker discovery approaches (e.g., logistic regression, LASSO, or previously reported ratio-based classifiers). We have therefore revised the Discussion to better articulate the rationale for introducing the UDR method. Specifically, we emphasize that UDR is a simple and interpretable ratio-based approach that minimizes the risk of overfitting in small cohorts and highlights the relative balance between up- and downregulated signals, which may be particularly suitable for clinical translation. At the same time, we have explicitly acknowledged in the Limitations section that further large-scale validation and head-to-head comparisons with traditional biomarker discovery methods will be required to confirm its robustness and clinical utility.
- The bioinformatics analyses, including target prediction and interaction networks, are described only superficially. The absence of critical details regarding databases, thresholds, and algorithms used has a detrimental effect on the interpretability and reproducibility of these results.
Response: In response to the reviewer’s concern, we have clarified the bioinformatics workflow in the revised manuscript. miRNA–target gene interactions were predicted and analyzed using the miRTargetLink 2.0 database in unidirectional search mode. To ensure high-confidence interactions, we applied filtration based on strong evidence and required a minimum of two shared targets. Network visualization was generated in concentric mode, and node highlighting was used to emphasize microRNAs and genes with higher connectivity or biological relevance. However, the specific algorithms used by the miRTargetLink 2.0 platform are not disclosed on the official website.
- In conclusion, although the manuscript addresses a clinically relevant topic and applies some innovative methodologies, it lacks sufficient methodological rigor, external validation, and in-depth biological exploration to support the robustness and generalizability of its findings. Future revisions should include more comprehensive patient cohort descriptions, independent validation in larger and more diverse populations, and deeper bioinformatics and functional analyses of the identified microRNAs. The study under review here constitutes an interesting preliminary exploration of the subject, but it does not yet reach the level of evidence required for a clinically actionable biomarker.
Response: We sincerely thank the reviewer for this thoughtful overall assessment. We fully acknowledge that our study represents an initial exploratory step toward establishing exosomal miRNA panels as diagnostic biomarkers for early-stage NSCLC. In the revised manuscript, we expanded the descriptions of patient selection, inclusion and exclusion criteria, and cohort characteristics to improve methodological clarity, and we added a dedicated Limitations section in the Discussion that explicitly addresses the restricted sample size, single-institution setting, absence of external validation, potential confounding factors, and lack of mechanistic validation while emphasizing the need for future large-scale multicenter studies. We also incorporated mechanistic evidence from prior literature into the Discussion to enhance the biological plausibility of the identified miRNAs. We agree that independent validation in larger and more diverse populations, as well as functional characterization of the identified miRNAs, will be essential before these biomarkers can be translated into clinical practice, and we view the present study as providing important preliminary evidence and a methodological framework that can guide and justify such future investigations.
Reviewer 2 Report
Comments and Suggestions for Authors
The aim of study was to identify exosomal microRNAs (miRNAs) panels for the non-invasive diagnosis of early-stage NSCLC. The next-generation sequencing was used in the discovery phase to profile 2,656 exosomal miRNAs in serum samples obtained from 76 patients with benign lung nodules and stage-specific NSCLC. The validation phase used qPCR to analyze 21selected miRNAs in serum samples obtained from 75 patients with benign lung nodules and stage-specific NSCLC. By use of the "up-to-down ratio" platform, a panel of four miRNA, including miR-150-5p, miR-301b-3p, miR-369-3p and miR-497-5p, was identified as diagnostic markers for early detection of lung cancer.
The four-phase approach used to identify the optimal miRNA panel for the non-invasive diagnosis of NSCLC is not novel as it has been reported by others. The sample size for the present study was much smaller than that by Yang L. et al. (PNAS. 2020;117 (40) 25036-25042), which identified the diagnostic miRNA panel based on the serum miRNA profiles obtained from 744 patients with NSCLC and 944 healthy controls. The miRNAs panel identified by Yang L. et al. consisted of let-7a-5p, miR-1-3p, miR-1291, miR-214-3p and miR-375. None of those miRNAs were included in the panel identified in this study. The authors should provide in depth discussions on the possible attributes of the discrepancy in miRNA panels among different studies.
What if the expression level of one of the four miRNAs is undetectable in the serum sample of a patient?
Author Response
The aim of study was to identify exosomal microRNAs (miRNAs) panels for the non-invasive diagnosis of early-stage NSCLC. The next-generation sequencing was used in the discovery phase to profile 2,656 exosomal miRNAs in serum samples obtained from 76 patients with benign lung nodules and stage-specific NSCLC. The validation phase used qPCR to analyze 21 selected miRNAs in serum samples obtained from 75 patients with benign lung nodules and stage-specific NSCLC. By use of the "up-to-down ratio" platform, a panel of four miRNA, including miR-150-5p, miR-301b-3p, miR-369-3p and miR-497-5p, was identified as diagnostic markers for early detection of lung cancer.
- The four-phase approach used to identify the optimal miRNA panel for the non-invasive diagnosis of NSCLC is not novel as it has been reported by others. The sample size for the present study was much smaller than that by Yang L. et al. (PNAS. 2020;117 (40) 25036-25042), which identified the diagnostic miRNA panel based on the serum miRNA profiles obtained from 744 patients with NSCLC and 944 healthy controls. The miRNAs panel identified by Yang L. et al. consisted of let-7a-5p, miR-1-3p, miR-1291, miR-214-3p and miR-375. None of those miRNAs were included in the panel identified in this study. The authors should provide in depth discussions on the possible attributes of the discrepancy in miRNA panels among different studies.
Response: We thank the reviewer for raising this important point. We agree that a four-phase approach has been previously applied by other groups, including the study by Yang et al. (PNAS 2020), and we have clarified that our novelty lies not in the overall study design but in the specific use of exosomal serum miRNAs and the development of the UDR platform as a simple ratio-based method to optimize panel performance. We also acknowledge the discrepancy between our identified panel and that of Yang et al., and we have now added a discussion noting that such differences likely arise from variations in sample sources (exosomal vs. total serum miRNAs), reference groups (benign pulmonary nodules vs. healthy controls), patient populations and clinical backgrounds, as well as analytical strategies. These factors, along with the relatively small sample size of our exploratory study, may account for the differences in miRNA composition across studies, and we have emphasized that larger, multi-cohort investigations will be required to reconcile these discrepancies.
- What if the expression level of one of the four miRNAs is undetectable in the serum sample of a patient?
Response: We thank the reviewer for this practical comment. In our dataset, all four selected miRNAs were consistently detectable above the assay threshold in every sample, and therefore this issue did not affect the construction or evaluation of our diagnostic panels. Nevertheless, we acknowledge that undetectable expression may occur in clinical practice. Because the UDR platform is ratio-based and uses the mean expression of multiple upregulated or downregulated miRNAs rather than relying on a single marker, the potential effect of one missing value would be minimized compared with single-miRNA assays. We have clarified this point in the Discussion and noted that further validation in larger and more diverse cohorts will be necessary to confirm assay robustness under real-world conditions.
References
- Wozniak, M.B.; Scelo, G.; Muller, D.C.; Mukeria, A.; Zaridze, D.; Brennan, P. Circulating microRNAs as non-invasive biomarkers for early detection of non-small-cell lung cancer. PloS one 2015, 10, e0125026.
- Arab, A.; Karimipoor, M.; Irani, S.; Kiani, A.; Zeinali, S.; Tafsiri, E.; Sheikhy, K. Potential circulating miRNA signature for early detection of NSCLC. Cancer Genetics 2017, 216, 150-158.
- Fehlmann, T.; Kahraman, M.; Ludwig, N.; Backes, C.; Galata, V.; Keller, V.; Geffers, L.; Mercaldo, N.; Hornung, D.; Weis, T. Evaluating the use of circulating microRNA profiles for lung cancer detection in symptomatic patients. JAMA oncology 2020, 6, 714-723.
- Ying, L.; Du, L.; Zou, R.; Shi, L.; Zhang, N.; Jin, J.; Xu, C.; Zhang, F.; Zhu, C.; Wu, J. Development of a serum miRNA panel for detection of early stage non-small cell lung cancer. Proceedings of the National Academy of Sciences 2020, 117, 25036-25042.
- Abdipourbozorgbaghi, M.; Vancura, A.; Radpour, R.; Haefliger, S. Circulating miRNA panels as a novel non-invasive diagnostic, prognostic, and potential predictive biomarkers in non-small cell lung cancer (NSCLC). British Journal of Cancer 2024, 131, 1350-1362.
- Dama, E.; Colangelo, T.; Cuttano, R.; Dziadziuszko, R.; Dandekar, T.; Widlak, P.; Rzyman, W.; Veronesi, G.; Bianchi, F. A plasma 9-microRNA signature for lung cancer early detection: a multicenter analysis. Biomarker Research 2025, 13, 74.